# SCN1B Genetic Variants: A Review of the Spectrum of Clinical Phenotypes and a Report of Early Myoclonic Encephalopathy

**DOI:** 10.3390/children9101507

**Published:** 2022-10-01

**Authors:** Zahra Zhu, Elizabeth Bolt, Kyra Newmaster, Wendy Osei-Bonsu, Stacey Cohen, Vishnu Anand Cuddapah, Siddharth Gupta, Sita Paudel, Debopam Samanta, Louis T. Dang, Paul R. Carney, Sunil Naik

**Affiliations:** 1College of Medicine, Penn State University, Hershey, PA 17033, USA; 2Epilepsy Neurogenetics Initiative, Division of Neurology, Children’s Hospital of Philadelphia, Philadelphia, PA 19104, USA; 3Kennedy Krieger Institute, Department of Neurology, Johns Hopkins University, Baltimore, MD 21218, USA; 4Department of Pediatrics and Neurology, Penn State Health Milton Hershey Medical Center, Hershey, PA 17033, USA; 5Division of Pediatric Neurology, Arkansas Children’s Hospital, Little Rock, AR 72202, USA; 6Department of Pediatrics, Division of Pediatric Neurology, University of Michigan Medical School, Ann Arbor, MI 48109, USA; 7Pediatric Neurology Division, University of Missouri Health Care, Columbia, MO 65212, USA

**Keywords:** SCN1B, voltage-gated sodium channel beta-1, Dravet-like syndrome, DEE, EME, fenfluramine

## Abstract

Background: Pathogenic variants in SCN1B, the gene encoding voltage-gated sodium channel b1/b1B subunits are associated with a spectrum of epileptic disorders. This study describes a child with early myoclonic encephalopathy and a compound heterozygous variant in the SCN1B gene (p.Arg85Cys and c.3G>C/p.Met1), along with the child’s clinical response to anti-seizure medications (ASMs) and the ketogenic diet. We reviewed the current clinical literature pertinent to SCN1B-related epilepsy. Methods: We described the evaluation and management of a patient with SCN1B-related developmental and epileptic encephalopathy (DEE). We used the Medline and Pubmed databases to review the various neurological manifestations associated with SCN1B genetic variants, and summarize the functional studies performed on SCN1B variants. Results: We identified 20 families and six individuals (including the index case described herein) reported to have SCN1B-related epilepsy. Individuals with monoallelic pathogenic variants in SCN1B often present with genetic epilepsy with febrile seizures plus (GEFS+), while those with biallelic pathogenic variants may present with developmental and epileptic encephalopathy (DEE). Individuals with DEE present with seizures of various semiologies (commonly myoclonic seizures) and status epilepticus at early infancy and are treated with various antiseizure medications. In our index case, adjunctive fenfluramine was started at 8 months of age at 0.2 mg/kg/day with gradual incremental increases to the final dose of 0.7 mg/kg/day over 5 weeks. Fenfluramine was effective in the treatment of seizures, resulting in a 50% reduction in myoclonic seizures, status epilepticus, and generalized tonic-clonic seizures, as well as a 70–90% reduction in focal seizures, with no significant adverse effects. Following the initiation of fenfluramine at eight months of age, there was also a 50% reduction in the rate of hospitalizations. Conclusions: SCN1B pathogenic variants cause epilepsy and neurodevelopmental impairment with variable expressivity and incomplete penetrance. The severity of disease is associated with the zygosity of the pathogenic variants. Biallelic variants in *SCN1B* can result in early myoclonic encephalopathy, and adjunctive treatment with fenfluramine may be an effective treatment for *SCN1B*-related DEE. Further research on the efficacy and safety of using newer ASMs, such as fenfluramine in patients under the age of 2 years is needed.

## 1. Background

Variants in genes encoding voltage-gated sodium channel (VGSC) a and b subunits are linked to epilepsies, including developmental and epileptic encephalopathies (DEE) [1,2,3]. Voltage-gated sodium channel b1/b1B subunits are encoded by the *SCN1B* gene [4]. Several pathogenic *SCN1B* genetic variants have been reported in individuals with DEEs [1,2] including Dravet-like syndrome [3], genetic epilepsy with febrile seizures plus (GEFS+) [5,6,7], and focal epilepsy [8]. *SCN1B* variants also cause cardiac arrhythmogenic conditions such as Brugada syndrome [9,10], and herein, we will focus on the neurodevelopmental and epileptic manifestations of *SCN1B* pathogenic variants. We describe a 16-month-old patient with early myoclonic encephalopathy (EME) attributable to compound heterozygous *SCN1B* genetic variants. EME is a rare disease characterized by neonatal or infantile onset myoclonic seizures [11]. Our study analyzes a unique case to support the linkage of *SCN1B* pathogenic variants to DEE. We also discuss the challenge of treating and managing patients with this phenotype and discuss the use of fenfluramine as a potentially effective treatment option.

### 1.1. Structure and Function of Voltage-Gated Sodium Channels

VGSCs are essential for the generation and propagation of action potentials during cellular depolarization throughout the central and peripheral nervous system and other organs, such as the heart [12]. VGSCs are heterotrimeric protein complexes, consisting of one central pore-forming a subunit, a covalently linked b subunit (b2 or b4) and a noncovalently linked b subunit (b1 or b3) [13] (Figure 1).

### 1.2. Voltage-Gated Sodium Channel b Subunits Encoded by SCN1B, SCN2B, SCN3B, and SCN4B

There are 4 VGSC beta subunit genes, *SCN1B, SCN2B, SCN3B*, and *SCN4B*, encoding 5 proteins, b1, b1B, b2, b3, and b4. b1 and b1B are splice isoforms encoded by the *SCN1B* gene. b1, b2, b3, and b4 contain an extracellular Ig domain, one transmembrane domain, and a regulatory intracellular domain with multiple phosphorylation sites and secretase cleavage sites [14,15,16,17,18]. In contrast, b1B is a developmentally regulated, secreted CAM that can function as a ligand to promote neurite outgrowth [19].

While the b subunits are not pore-forming, they regulate VGSC localization and kinetics as well as participate in cell–cell adhesion and regulated intramembrane proteolysis as immunoglobulin (Ig) superfamily cell adhesion molecules (CAMs) [13,20,21]. The effect of b1 expression on the gating and kinetics of the VGSC pore has been studied in heterologous expression systems. In *Xenopus* oocytes and Chinese hamster lung cells, coexpression of the alpha subunit Na_v_1.2 with b1 resulted in faster sodium current activation and inactivation and a higher peak current than expression of Na_v_1.2 alone [16,22]. b1 also results in increased a subunit cell-surface expression [23]. b1 can also modulate the kinetics and cell-surface expression of voltage-gated potassium channels [24,25,26], however these interactions have not been fully explored to the extent of VGSCs. *Scn1b*-null mice that lack both b1 and b1B exhibit spontaneous seizures, ataxia, poor weight gain, and premature death around post-natal day 19, modeling Dravet Syndrome, a DEE [27]. Patch clamp recordings from these mice displayed neuronal cell-type, brain region-specific, and cortical layer-specific abnormalities in neuronal function [22,27,28].

In addition to its function in modulating the kinetics and expression of the a subunit, b1 also functions as a CAM through its extracellular Ig domain [29] and can participate in *trans* homophilic cell adhesion when expressed in *Drosophila* S2 cells, resulting in downstream signaling events [30,31]. b1 can also bind to a variety of other CAMs.

## 2. Methods

Consent was obtained from the patient’s family to publish a case report. A literature review was performed by searching for “SCN1B” and “epilepsy” in Medline and Pubmed. Relevant articles describing SCN1B-related epilepsy were selected manually, and genotypic, phenotypic, and functional data were extracted for analysis.

## 3. Results

### 3.1. Case Report

A 16-month-old male diagnosed with early infantile epileptic encephalopathy (DEE), Dravet-like syndrome and global developmental delay was found to have biallelic *SCN1B* genetic variants. He was born full-term via an induced vaginal delivery due to gestational hypertension and had no perinatal complications. Since three weeks of age, he started having staring episodes with extremity twitching. At that time, electroencephalography (EEG) was reported as normal.

Two months later, his extremity jerking increased in frequency. At times, these myoclonic movements were nearly continuous but did not interfere with his playfulness. An MRI of the brain at 2 months of age was unremarkable, except for incomplete myelination of the anterior limb of the internal capsule and the corpus callosum. At four months of age, he presented to the hospital with convulsive status epilepticus. He received and responded to benzodiazepines and was started on levetiracetam (30 mg/kg/day). A comprehensive neurometabolic workup was performed at that time, including an Invitae epilepsy gene panel testing. This panel included 320 add-on preliminary genes to test for genetic disorders associated with epilepsy [32].

A few weeks later, he presented to the hospital at that time with another episode of possible status epilepticus. Neurological examination showed global hypotonia and hyperreflexia with nearly continuous myoclonus. He was given benzodiazepines and levetiracetam at a high dose (163 mg/kg/day). A continuous EEG (cEEG) revealed occasional sharp discharges in the bilateral occipital regions. Multiple episodes of his myoclonic jerking were captured on cEEG and the findings were suggestive of both myoclonic seizures and subcortical nonepileptic myoclonus. He had further similar inpatient admissions for status epilepticus with no meaningful response to levetiracetam (163 mg/kg/day), topiramate (12 mg/kg/day), pyridoxine (100 mg intravenously × 3 doses) followed by maintenance dose (30 mg/kg/day), clobazam (2.6 mg/kg/day), phenobarbital (5.2 mg/kg/day), clonazepam and a classic ketogenic diet (beta-hydroxybutyrate level/BHB > 3–4 mmol/L) with good adherence. He developed significant somnolence due to the various combinations of anti-seizure medications (ASM).

An epilepsy gene panel revealed two *SCN1B* variants in *trans* (compound heterozygous with NM_199037.3; NM_001037.4 c.253C>T; p.Arg85Cys and c.3G>C; p.Met1?). The SCN1B p.Arg85Cys variant was in a heterozygous state, paternally inherited, and classified as likely pathogenic (ClinVar variation ID 190859). SCN1B p.Arg85Cys variant has been described as pathogenic in the literature associated with infantile intractable epilepsy [1]. The SCN1B p.Met1? variant was in a heterozygous state, maternally inherited, and classified as a variant of uncertain significance originally (ClinVar variation ID 1040220), but this was upgraded to likely pathogenic by the diagnostic laboratory due to increased frequency of the variant in affected individuals.

There were additional variants found in LMNB2, RBFOX3, RELN and SCP2 genes. These variants were deemed as Variants of Uncertain Significance (VUS) by Invitae lab that adheres closely to the recommendations by American College of Medical Genetics (ACMG) for variant classification [33]. VUSs are genetic variations about which sufficient clinical or functional evidence is unavailable in order to be classified as definitively pathogenic or benign. Hence, these VUSs were thought to not contribute to the disease phenotype in our patient.

His parents, who were heterozygous carriers, had no history of seizures or other neurodevelopmental disorders. Family history is otherwise unremarkable. His only sibling was a full biologically healthy 4-year-old brother with normal development and with no neurological concerns. His metabolic workup, Brain MRI and magnetic resonance spectroscopy (MRS) results were negative. Due to his genotype and correlating phenotype, he was given the diagnosis of *SCN1B*-related early myoclonic encephalopathy with probable Dravet-like Syndrome.

The majority of his hospitalizations were due to status epilepticus without identified triggers such as infections or stressors. Repeat cEEG revealed abnormal and worsening of the background findings during each admission. Multifocal abundant sharp wave, spike and wave complexes, generalized polyspike and spike and wave discharges were seen. Occasional to nearly continuous clusters of myoclonic seizures with time-locked EEG changes (spike and wave complexes) were captured. Interestingly, many of the myoclonic jerks were also not associated with any EEG changes and it was often difficult to clinically differentiate between cortical and subcortical myoclonus. Ophthalmology examination did not reveal intraocular stigmata, ruling out various neurodegenerative etiologies. The patient was not able to tolerate pyridoxine and pyridoxal 5′-phosphate (both at 30 mg/kg/day), hence it was discontinued after a negative metabolic and genetic work for pyridoxine-dependent epilepsy (Figure 2).

The patient also displayed severe psycho-motor regression including excessive somnolence, poor eye contact, loss of a social smile, cooing and neck control likely due to prolonged seizures and the use of multiple ASMs. Significant distress was noticed in association with continuous multifocal axial and extremity myoclonus, even after resolution of status epilepticus. This resulted in the frequent usage of benzodiazepines in the ICU. A gastrostomy tube was placed due to poor oral feeding. The patient remained admitted to the hospital due to respiratory insufficiency secondary to infections and worsening of seizures.

Due to a failure to control episodes of status epilepticus, near continuous debilitating cortical and subcortical myoclonus, repeated inpatient and ICU admissions, fenfluramine was discussed as a potential medication option with his parents. A formal cardiology consultation was ordered and initial cardiology workup, including serial EKGs and echocardiogram (ECHO) were negative, except for a trivial patent foramen ovale (PFO) with left to right shunting.

Fenfluramine was started at 8 months of age at 0.2 mg/kg/day with gradual incremental increases to the final dose of 0.7 mg/kg/day. This transition was performed over five weeks without complications or side effects. Serial follow-up was performed every few months as recommended by the cardiology team. The first ECHO after starting fenfluramine revealed an estimated peak right ventricular systolic pressure of 32 mmHg, which is consistent with borderline elevated pressure. Fenfluramine was continued and the next ECHO was performed in 3 months, revealing the average pulmonary artery acceleration time (PAAT) to the right ventricular ejection time (RVET) ratio to be 0.36. This implies the absence of pulmonary hypertension [34]. As per cardiology recommendations, the patient was recommended to avoid all drugs contraindicated for patients with Brugada syndrome [35].

During the next 3 months, the patient tolerated fenfluramine well without any significant side effects. A gradual reduction in myoclonus was reported by parents starting as early as the first month of initiating fenfluramine (Table 1). Clobazam and topiramate had been weaned off without a worsening of his seizures (Figure 2). In addition, he had a reduction in the frequency of his status epilepticus, and an improvement of his baseline myoclonic and GTC seizures (Table 1).

The parents were taught by pediatric neurologists to recognize the various types of seizures that the patient was affected with. Seizure burden (number, duration and type of seizures) were quantified, and a record was kept by both the parents and the visiting home health nursing team, in addition to the epilepsy nurse navigator from the epilepsy clinical program. Although these records were subjective, the parents were educated about seizure types and how to document this information. Parents and nursing team also recorded short videos of the seizures and sent it to the pediatric neurology epilepsy team for confirmation if they were unsure if the episode was a seizure. We felt clinically that the parent’s reporting of seizures was very reliable, as they were closely monitoring the patient’s seizures for the last 13 months. The percentage of seizure reductions reflect our best clinical judgement of the seizure burden affected by the patient.

The parents and nursing team were also able to abort most of his seizure clusters at home by serial use of benzodiazepines as needed. He had 2 more brief admissions due to respiratory failure secondary to respiratory infections, resulting in the worsening of his seizures. Hence, valproate (54 mg/kg/day), phenobarbital (5.2 mg/kg/day), and cannabidiol (Epidiolex^R^, 21 mg/kg/day) were added during the next 4 months. He did not have any episodes of status epilepticus for the 3 months on a regimen of levetiracetam (163 mg/kg/day), fenfluramine (0.7 mg/kg/day), valproate, cannabidiol, phenobarbital, and the ketogenic diet. His parents also reported improved alertness and improved truncal tone, and he achieved new motor milestones.

At 18 months of age, the patient was tolerating fenfluramine and other ASMs with the KD and had serial follow-up with cardiology with a negative cardiac workup as an outpatient. Of note, the patient did not have any reported cardiac arrhythmias on telemetry in ICU settings, despite prolonged status epilepticus and the use of fenfluramine.

### 3.2. Review of the Spectrum of Associated Phenotypes with SCN1B Variants

Several monoallelic variants in *SCN1B* have been associated with febrile seizures, febrile seizures plus, GEFS+, early-onset absence epilepsy, and focal epilepsy including temporal lobe epilepsy (TLE), and patients who experience sudden unexpected death from epilepsy (SUDEP) (Table 2). Biallelic variants in *SCN1B* are associated with DEE, ranging from severe early onset DEE to a phenotype milder than but resembling Dravet syndrome (Table 2). Most variants are missense, with a couple splice site variants, and all affect the extracellular domain of b1 or b1B. Functional studies on many of the missense variants have been performed using heterologous expression systems or a mouse model, and the results of select studies are summarized in Table 2. As shown in Table 2 below, only 1 variant (C121W) was tested using an animal model. All other variants were tested in vitro. With few variants having been tested in vivo, in vitro data is largely unsupported in terms of loss of function/gain of function. We acknowledge this as a limitation.

#### 3.2.1. Genetic Epilepsy with Febrile Seizures Plus (GEFS+)

GEFS+ is a genetically heterogeneous familial syndrome with autosomal dominant inheritance, characterized by a predominance of febrile seizures (FS) among other seizure types of variable phenotypic severity. It may be the initial presentation for various epilepsy syndromes, such as early myoclonic encephalopathy and Dravet syndrome. The most typical presentations include FS, febrile seizures (+) namely those that occur after 6 years of age, and afebrile generalized tonic-clonic seizures.

There are also other uncommon manifestations of GEFS+ including myoclonic, absence, and atonic seizures [5,6,44]. Severe phenotypes include myoclonic-astatic epilepsy and GEFS+ can be on the spectrum with Dravet Syndrome (formerly referred to as severe myoclonic epilepsy of infancy).

*SCN1B* p.C121W was the first genetic variant found to be associated with GEFS+ (OMIM 604233) and results in a disruption of the disulfide bond that is needed to form the Ig loop [48]. Additional heterozygous mutations in the *SCN1B* gene including p.Met1?, p.D25N, p.I70-E74del, p.R85C, p.R85H, and p.R125L have been also found to result in GEFS+. Candidate pathogenic variants p.R89H and p.R96Q are possibly associated with epilepsy, although these were single case descriptions without demonstration of segregation of phenotypes with the genotypes in a larger pedigree [41,42].

The disease penetrance of *SCN1B* p.C121W and p.I70_E74del mutations for GEFS+ is found to be 62–76% [5,6,44].There is also variable expressivity, and family members that share the same *SCN1B* variant may be seizure-free, with febrile seizures, or with epilepsy (most frequently FS+). Overall, variants of the SCN1B gene have variable expressions, impacting both excitatory and inhibitory neurons, affecting the severity of the clinical expression. Moreover, loss of function genetic variants may result in a gain of function mechanism, producing hyperexcitability.

#### 3.2.2. Focal Epilepsy

TLE is characterized by recurrent, focal seizures, often occurring in clusters, and are typically associated with an aura of familiarity or unfamiliarity, loss of awareness, and oral automatisms.

Scheffer et al. identified families with mutations in the beta-1 subunit of the voltage-gated sodium channel and characterized the spectrum of phenotypes [8]. TLE was among the spectrum of clinical manifestations with a *SCN1B* p.C121W, along with GEFS+, and FS [9]. Some of the individuals with TLE had preceding convulsive FS that can secondarily cause hippocampal sclerosis and TLE, but at least one individual did not have convulsive FS prior to the onset of TLE. This implies that SCN1B variants may directly result in TLE although a long term follow up is necessary in the future to support this correlation.

Another case of focal epilepsy was associated with a p.D25N variant, notably the only de novo *SCN1B* variant previously described [37]. It was unclear whether the focal epilepsy in this individual was localized to the temporal lobes. Functional studies of the p.D25N variant suggested a loss-of-function (LOF), similar to functional studies on other *SCN1B* variants.

#### 3.2.3. SCN1B-Related Developmental and Epileptic Encephalopathy

Patients with biallelic SCN1B variants have been reported in the literature with clinical manifestations consistent with or sharing features of Dravet Syndrome (DS) [3,49], a developmental and epileptic encephalopathy [50]. In DS, seizures usually begin around 6 to 12 months of age, with prolonged febrile, generalized, clonic or hemi-clonic seizures. EEG has a normal background until the first few years of age. After 2 years of age, individuals with DS may begin showing signs of developmental decline. While its clinical presentation is variable, many individuals with DS have psychomotor regression along with intractable epilepsy [51]. Other neuropsychiatric symptoms, such as gait abnormalities, sleep disorders, and behavioral and cognitive changes present in the next few years of life.

The diagnosis of DS is based on seizure types, age of onset and clinical course of symptoms and is supported by the presence of an *SCN1A* mutation, and other diagnostic testing [52]. Identification of pathogenic *SCN1A* variant is not mandatory for Dravet syndrome diagnosis. Dravet syndrome should be clinically suspected if a child has most of these following: (1) normal development before the first seizure, (2) two or more seizures, with or without fever, before age 1, (3) two or more seizures that last more than 10 min, (4) myoclonic, hemi-clonic or tonic-clonic seizures, and/or (5) seizures that do not respond to epilepsy medications, with seizures continuing past age 2. Although a large majority of patients with Dravet syndrome have an *SCN1A* gene mutation [53], approximately 20% of the patients do not have an identified pathogenic variant in *SCN1A*. Therefore, researchers looked for other genetic etiologies, such as an association between Dravet Syndrome and mutations in the *SCN1B* gene.

Ogiwara et al. studied 67 patients with Dravet Syndrome, who were negative for both SCN1A and SCN2A mutations [3]. In one patient, they discovered a novel homozygous mutation in the *SCN1B* gene (p.Ile106Phe) in which a single nucleotide change occurred, leading to an amino acid substitution [3]. This missense mutation, which resulted in isoleucine being replaced by a more bulky phenylalanine, is thought to disrupt the interaction between the VGSC beta-1 subunit and cellular adhesion molecules, leading to neuronal hyperexcitability and epilepsy [3]. The patient had recurrent hemi-clonic and myoclonic seizures by the age of 6 months, followed by atypical absence seizures and status epilepticus at 12 and 13 months, respectively [3]. The patient also showed developmental delays by the age of 4 and was suffering from general tonic-clonic seizures more than once a week [3].

Similarly, in a study of 5 children from 3 different families with DEE, Ramadan et al. discovered 2 novel recessive SCN1B variants in children with early onset epilepsy and severe developmental delays [43]. Although dominant *SCN1B* variants are known to be associated with childhood epilepsy syndromes, this study confirms that recessive variants must also be considered in children with Dravet-like phenotypes [41].

Dravet syndrome is also known to be significantly associated with sudden unexpected death in epilepsy (SUDEP). A *Scn1b*-null mouse model mimics a Dravet-like phenotype characterized by severe, early onset seizures and lethality by postnatal day 21 [27]. As the beta-1 subunits encoded by *SCN1B* genes are Ig-CAMs and play a role in neural development, it is crucial to understand if abnormal neural development causes subsequent epileptic activity. O’Malley et al. used developmentally normal, conditional knock-out (cKO) mice to delete *Scn1b* in adult mice to test the hypothesis that abnormal neurodevelopment is necessary for epileptogenesis [54]. In mice that developed normally, conditional KO of *Scn1b* still resulted in severe epilepsy and SUDEP, suggesting that aberrant neuronal development does not underlie *Scn1b*-related epileptogenesis [52]. Although abnormal brain development is a characteristic of SCN1B-associated DEE, severe epilepsy and SUDEP may not stem from the underlying impaired neurodevelopment [52].

Developmental and epileptic encephalopathy, 52 (DEE52) is a rare, severe disorder affecting neurodevelopment that begins in infancy, presenting with intractable seizures and developmental stagnation or regression (OMIM 617350). DEE52 is linked to an autosomal recessive, biallelic, loss-of-function mutation in the *SCN1B* gene. Although *SCN1B* variants have previously been linked to Dravet Syndrome, Aeby et al. propose that DEE is better suited to characterize patients with biallelic *SCN1B* variants [1]. A female patient with hypotonia and abnormal development at birth as well as epilepsy by the age of 3 months was found to have the homozygous *SCN1B* variant p.Arg85Cys [1]. Due to loss of the ability to modulate Na_v_1.1-generated sodium current, this patient had a more severe phenotype than typical Dravet Syndrome patients, with an abnormal EEG at 3 months and an earlier onset of seizures and developmental delay [2].

In a study of nine patients from four unrelated families with three different biallelic variants of *SCN1B*, functional studies revealed that these *SCN1B* missense variants have the potential to alter voltage-gated sodium channel gating kinetics, likely contributing to the development and severity of DEE52 [2].

### 3.3. Review of the Medications Reported to Be Beneficial in SCN1B-Related Disorders

#### 3.3.1. Phenytoin in SCN1B-Related Refractory Epilepsy

The type of SCN1B genetic mutation may be helpful in predicting which anti-seizure medication would be helpful. A paternally inherited p.R89H monoallelic variant in SCN1B was described in a case of a 2-year-old patient with refractory epilepsy. As the patient continued to have seizures despite taking valproate, levetiracetam, topiramate and then clobazam, phenytoin was subsequently initiated at age 4. He had dramatic improvement in his seizures and was able to wean all other anti-seizure medications over next 2 years. Sodium channel blockers may be effective in individuals with certain SCN1B-related epilepsies, however further studies are necessary to bolster this finding. p.R89H variants of *SCN1B* gene may result in a gain of function for the alpha subunit of the VGSC due to the loss of the modulating function of beta-1. Sodium channel blockers may be effective in these cases.

#### 3.3.2. Fenfluramine in Dravet-like Syndrome

Fenfluramine is an FDA-approved medication for patients over the age of 2 with seizures from Dravet syndrome or Lennox-Gastaut Syndrome [55]. It is an amphetamine derivative that modulates serotonin neurotransmitter levels [56]. Fenfluramine’s three postulated mechanisms for modulating serotonin include increasing the release of serotonin, inhibiting serotonin’s reuptake and directly acting on the serotonin receptors [54]. Fenfluramine’s anti-seizure properties were elucidated in animal models 20 years before fenfluramine was first used in the 1970s in adults to treat obesity. In the 1980s, it was used to modify behavior in autistic patients. It was also noted to have significant seizure reduction effects in these patients. It is a serotonergic agent that was used by millions of patients, however it was removed from the market in the late 1990s after it was found to be associated with serious cardiovascular adverse events, such as cardiac valvulopathy and pulmonary hypertension [57].

When started in a 28-month-old patient with homozygous *SCN1B*-associated early infantile DEE, fenfluramine led to a cessation of fever-induced status epilepticus and a reduction in seizure frequency and severity. Although there was no improvement in motor or cognitive development following the addition of fenfluramine, this patient did not exhibit any adverse cardiovascular effects [1]. Considering *SCN1B* mutations may be associated with cardiac arrhythmic disorders, the potential use of fenfluramine must involve a cautious decision involving weighing the benefits and potential harm.

Serial retrospective and double-blinded studies were conducted in 2012–2020 [58,59,60,61] in patients with Dravet syndrome associated with *SCN1A* with results showing excellent efficacy and tolerance. Moreover, there were no cardiac complications, although most of the patients had no underlying cardiac problems. It was approved for use in the treatment of seizures associated with *SCN1A* Dravet syndrome (and subsequently for Lennox-Gastaut Syndrome) in patients 2 years of age and older with a warning of valvular heart disease and pulmonary hypertension. Therefore, the use of this medication requires thorough cardiac testing prior to and during fenfluramine use.

Due to severe, intractable, progressive myoclonic epilepsy in the index case described herein, numerous anti-seizure medications were tried without significant seizure control. At 8 months old, he was started on fenfluramine in addition to his existing regimen of anti-seizure medication for better seizure control. This medication was effective for improved seizure control in this case in an infant with an *SCN1B* mutation, when used off-label from the FDA approved indications.

## 4. Discussion

Pathogenic *SCN1B* variants have previously been associated with generalized epilepsy with febrile seizures plus, Dravet syndrome, early infantile epileptic encephalopathy, inherited arrhythmia disorders such as Brugada syndrome and various other cardiac abnormalities.

We report this unique case of a patient with an SCN1B-related early myoclonic encephalopathy, presenting with debilitating cortical and subcortical myoclonus and prolonged status epilepticus with severe psychomotor regression. He also had numerous inpatient hospitalizations and refractory seizures to multiple ASM and the ketogenic diet. Both parents were also found to be the carriers for the variants. This report expands the genotypic spectrum by describing the first case of a compound heterozygous variant in *SCN1B* causing DEE, and expands the phenotypic spectrum by including EME as a manifestation of *SCN1B*-related epilepsy.

Fenfluramine, and later cannabidiol was added on compassionate basis as it does not have an FDA approval for use in patients less than 1 year of age and in Dravet syndrome associated with the SCN1B genotype–phenotype. This patient was started on fenfluramine on a compassionate basis after parental approval and cardiology evaluation and approval. As described above, significant improvement of debilitating myoclonus was seen in addition to reduced attacks of intractable status epilepticus. The patient also tolerated other ASM such as high dose levetiracetam, cannabidiol, phenobarbital and valproic acid. This combination of ASMs reduced hospitalization (none over last 3 months at the time of the reporting) and seizure frequency. No abnormal cardiac findngs occurred, despite the theoretical risk of arrhythmias associated with Brugada syndrome in *SCN1B*-related phenotypes.

As per parental and various subspecialist reporting, we found that fenfluramine was well tolerated and effective in treating the various types of seizures in our patient at less than 1 year of age with *SCN1B*-related DEE. He continues to tolerate fenfluramine and other ASMs well without any significant adverse effects 8 months from initiation of fenfluramine.

## 5. Conclusions and Future Directions

A multidisciplinary evaluation of large sample sizes will help to recognize various seizures semiology, psychomotor retardation, co-existent cardiac arrythmias and risk of SUDEP in patients with *SCN1B*-associated early myoclonic encephalopathy. Extrapolation from studies in children >2 years of age may be sufficient for clinical usage and observational studies on the usage of fenfluramine and other ASMs in young infants with *SCN1B*-related DEE, as double-blinded randomized controlled multi-centered studies are unlikely to be feasible with such a small patient population. More research on which types of *SCN1B* variants would benefit from specific ASMS would also be helpful for clinical practice. If found to be safe and efficacious in *SCN1B*-associated DEE, fenfluramine could be used in early infancy to reduce incidence of developmental disability, status epilepticus and prolonged hospitalizations.

## Figures and Tables

**Figure 1 children-09-01507-f001:**
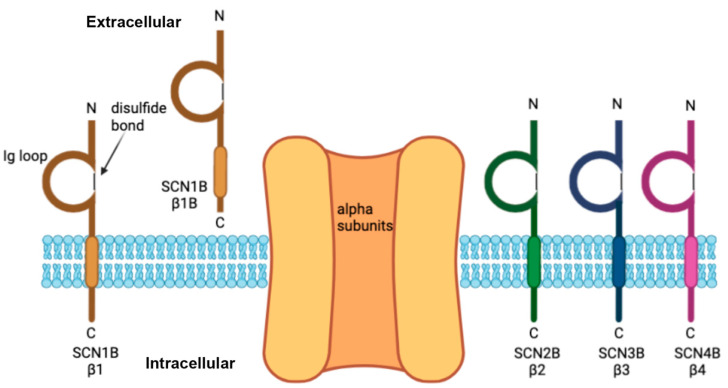
Schematic of the voltage-gated sodium channel b subunits encoded by *SCN1B*, *SCN2B*, *SCN3B*, and *SCN4B*. The b subunits associate with an alpha (pore-forming) subunit and have an extracellular immunoglobulin (Ig) loop with a single transmembrane domain. b1B is a secreted cell adhesion molecule.

**Figure 2 children-09-01507-f002:**
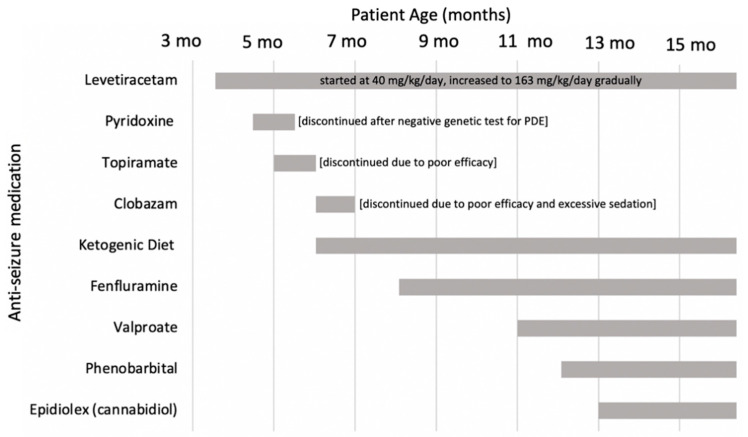
Timeline of initiating and/or discontinuing anti-seizure medications. PDE = pyridoxine-dependent epilepsy.

**Table 1 children-09-01507-t001:** Clinical response to anti-seizure medications.

	Levetiracetam/Clobazam, Topiramate	Fenfluramine	Epidiolex (Cannabidiol)	Valproic Acid/Phenobarbital
MYOCLONIC SEIZURE	No improvement	50% reduction	60% reduction	90% reduction
STATUS EPILEPTICUS	No improvement	50% reduction	60% reduction	90% reduction
FOCAL SEIZURES	No improvement	70–90% reduction		
GTC	No improvement	50% reduction	60% reduction	60% reduction
HOSPITALIZATIONS	No improvement	50% reduction	70% reduction	90% reduction

**Table 2 children-09-01507-t002:** Summary of SCN1B variants causing epilepsy and neurodevelopmental delay, with functional studies on specific variants.

Variant in SCN1B (NM_001037.5)	Protein Change	Zygosity	Phenotype	Number of Families or Individuals	Clinical Reference	Summary of Functional Studies
c.1A>T	p.Met1?	Het	FS, GEFS+, SUDEP	1 family (6 individuals)	Myers et al. [36]	None
c.73G>A	p.D25N (de novo)	Het	Focal epilepsy	1 individual	Orrico et al. [37]	Decreased targeting to plasma membrane, does not modulate gating of alpha subunit. LOF. [38]
c.136C>T	p.R46C	Homozygous	Developmental and epileptic encephalopathy 52 (DEE52)	1 family	Scala M et al. [2]	In Xenopus oocyes, co-expression of p.R46C with various alpha subunits resulted in shift toward more depolarized potentials in the conductance-voltage relationship (with Na_v_1.2 and Na_v_1.6) and in channel availability (with Na_v_1.1 and Na_v_1.2) compared to wild-type b1. [2]
c.178C>T	p.R60C	Homozygous	Developmental and epileptic encephalopathy 52 (DEE52)	1 family	Scala M et al. [2]	In Xenopus oocyes, co-expression of p.R60C with various alpha subunits resulted in shift toward more depolarized potentials in the conductance-voltage relationship (with Na_v_1.2 and Na_v_1.6) and in channel availability (with Na_v_1.2) compared to wild-type b1. [2]
IVS2-2AC	p.I70_E74del	Het	FS, GEFS+ and early-onset absence, penetrance 63%	1 family	Audenaert et al. [5]	None
c.253C>T	p.R85C	Homozygous	DEE	1 individual in a family	Aeby et al. [1]	p.R85C cannot modulate Nav1.2 kinetics. LOF. [27] p.R85C expressed on cell surface, but did not modulate Nav1.1 kinetics. LOF. [1]
		Het	GEFS+	1 family	Scheffer et al. [8]	
			FS	1 family (2 individuals)	Aeby et al. [1]	
	p.R85H	Het	GEFS+	1 family	Scheffer et al. [8]	p. R85H did not modulate fast kinetics but did modulate slow inactivation of Nav1.2 like wild-type beta-1 does. Neither expressed on cell surface. Loss of function (LOF). [39]
c.265C>T	p.R89C	Homozygous	DEE, Dravet-like but more mild	1 family (2 siblings)	Darras et al. [40]	None
c.266G>A	p.R89H (Candidate pathogenic variant)	Het	Epilepsy with myoclonic-atonic seizures	1 individual	Dang et al. [41]	None
	p.R96Q (Candidate pathogenic variant)	Het	SUDEP, epilepsy	1 individual	Bagnall et al. [42]	None
	p.I106F	Homozygous	Dravet	1 individual	Ogiwara et al. [3]	None
c.355T>G	p.Y119D	Homozygous	DEE, microcephaly, severe developmental delay, periventricular leukomalacia, brain atrophy	1 family	Ramadan et al. [43]	None
c.363C>G	p.C121W	Het	GEFS+, penetrance 62%	1 family, 6 generations, 26 of which had GEFS+	Wallace et al. [44]	In Xenopus oocytes, co-expressed with alpha subunit, beta-1 p.C121W cannot modulate channel-gating kinetics. LOF. [44] p.C121W variant increases Na channel availability and reduces channel rundown (a gradual decrease in sodium current with high frequency activity) but does not affect sodium current kinetics or voltage-dependence of channel activation. Additionally, p.C121W disrupts cell adhesion function. LOF. [45] In HEK cells, beta4 increases persistent sodium current of Nav1.1, and beta1 counteracts the effect of beta 4. p.C121W beta1 variant is unable to counteract effect of beta 4. LOF. [46] Mice heterozyous for SCN1B p.C121W were more susceptible to hyperthermia-induced seizures than heterzygous SCN1B null mice. This implies a deleterious gain of function. [47]
			GEFS+, penetrance 76%	1 family, 19 individuals with epilepsy	Wallace et al. [6]	
			GEFS+, TLE	2 families	Scheffer et al. [8]	
	p.R125C	Homozygous	Dravet	1 individual	Patino et al. [22]	No cell surface expression in heterologous system resulting in LOF. [21]
c. 374C>T	p.R125L	Het	GEFS+	1 family	Fendri-Kriaa et al. [7]	None
c.449-2A>G	splice variant	Homozygous	DEE, severe developmental delay, early death, brain atrophy, neonatal onset myoclonic and tonic clonic epilepsy	2 families	Ramadan et al. [43]	None
c.472G>A	p.V158M	Homozygous	Developmental and epileptic encephalopathy 52 (DEE52)	2 families	Scala et al. [2]	In Xenopus oocyes, co-expression of p.V158C with various alpha subunits resulted in shift toward more depolarized potentials in the conductance-voltage relationship (with Na_v_1.6) and in channel availability with Na_v_1.1 and Na_v_1.6) and slower recovery from fast inactivation(with Na_v_1.1) compared to wild-type b1. [2]
	p.G257R of beta1B	Het	idiopathic epilepsy	2 families	Patino et al. [19]	Affects beta-1B, not beta1 protein, and leads to intracellular retention of beta-1B, which may affect neurite outgrowth during development. LOF. [18]

## Data Availability

Not applicable.

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
