# Peer review of "SCN1B Genetic Variants: A Review of the Spectrum of Clinical Phenotypes and a Report of Early Myoclonic Encephalopathy"

_children, 2022, doi:10.3390/children9101507_

Round 1

Reviewer 1 Report

Abstract lines 25-28: It is unclear whether the results summarized are from the literature or from the case study.

Introduction: Lines 62-63: Statement regarding B1B effect on neurite outgrowth requites a citation.

Introduction Line 108-109: Statement reads: “the heterozygous SCN1B C121W (SCN1B +/C121W) mice have a lower seizure threshold than heterozygous SCN1B null (SCN1B+/-) mice, and this implies a deleterious gain of function mechanism.” This is also included in Table 1. However, this implication is not completely supported by the data cited since the heterozygous missense variant should be compared to wild type littermates, and the same with the heterozygous null. It is unclear how a valid comparison was made between null hets and variant hets, unless the null and variant hets were crossbred, and then reassorted randomly with WT littermates.  Short of this, the support for in vivo gain of function for C121W is weak.

Introduction: Related to the above comment, only 1 variant (C121W) was tested using a mouse/animal model, all others were tested in vitro. This should be noted as a limitation, since in vivo and in vitro studies often contradict one another.

Table 1: Why is there no LOF/GOF designation for the last 3 entries?

Introduction lines 129-130: Statement reads: “This implies that SCN1B variants may directly result in TLE.” Were patients in this study screened for variants in other sodium channel genes? If not, this limitation should be mentioned since gene/protein interactions may be involved in such phenotypes. Similarly with other studies that are cited in regards to pathogenic effects of SCN1B variants in different epilepsy syndromes: were other sodium channel genes evaluated in those studies? If not, again this can be cited as a limitation.

Lines 208-210: Statement reads: “It is reported that various SCN1B-related developmental and epileptic encephalopathies impact both excitatory and inhibitory neurons, resulting in the increased severity of EIEE52.” This does not seem accurate and/or is confusing. Wouldn’t it be the variants that affect both types of neurons and result in increased severity (not the encephalopathies)? Related to this, wouldn’t most/all SCN1B (and SCN1A) variants cause sodium channel dysfunction in both excitatory and inhibitory neurons? There is also the issue that loss of function of the beta subunit may result in gain of overall channel function (i.e. hyperexcitability). These effects should be distinguished as they are highly relevant to therapeutic strategies. Overall more discussion/explanation of gain vs loss of function effects of variants would strengthen the manuscript.

Lines 228_230 regarding fenfluramine: “It is an amphetamine derivative that primarily affects serotonin neurotransmitter levels”. In fact, although synaptic levels of serotonin may be increased transiently (not overall brain levels), the statement as written disregards the changes in receptor density that accompany the therapeutic use of this drug and that are thought to underlie therapeutic benefits in depression. There are other effects as well, and these all should be briefly reviewed. Special attention should be given to downstream effects such as on global gene transcription and cellular effects such as neurogenesis.

Line 276: More information on the initial genetic panel should be provided.

Comments on the Case Report:

Lines 301-310: It would be interesting to examine possible association between medication changes and EEG changes. Was the patient on different combinations of drugs at the times of the various EEG tests?

Lines 321-322: “Early infancy age” does not seem like a criterion that would support or justify the use of a medication approved only for older children. It seems like the opposite.

Lines 324-325: How was it determined that the patent foramen ovale was not clinically significant or even related to recurring seizures?

Line 335: The statement is made regarding recommendations from cardiology for this patient: “..avoid all drugs contraindicated for patients with Brugada syndrome”. Did this include any drugs that the patient was taking at the time?

Table 2: The data in this table are not presented clearly. Thus, the therapeutic impact of the various treatments, including fenfluramine, cannot be determined. What drugs was the patient on when fenfluramine started? What was the time course of changes in the patient’s ASM regimen after fenfluramine was added? What was the order of weaning? What are the “% reductions” in reference to? Are these subjective? The authors should consider an alternative presentation for Table 2, with addition of more data.

What level of genetic testing and what other genetic information is available on the patient? Have other ion channel genes been screened?

Does the patient have any siblings?

Minor comments:

Throughout: English grammar requires refinement (e.g. lines 71-73 and 78-79).

There are instances where “patient-first” language is not used (e.g. line 83 should be: “patients who experience sudden death from epilepsy (SUDEP)”.

TLE is defined on line 121 but temporal lobe epilepsy is first mentioned on Line 85 and the acronym should be defined there. Also, the acronym should be used on line 125.

Lines 130-131: There seems to be word missing in this sentence. Should this be: “..notably the only de novo SCN1B [VARIANT] previously described”?

LOF is define on Line 134, but is used several times before this including in Table 1. Similar problems are noted with other acronyms as well (e.g. ECHO). Poor organization and lack of systematic use of acronyms detracts from the manuscript.

Lines 142-143: This is not a complete sentence

Note redundancy between lines 244-245 and 251-252 regarding need for cardiac monitoring during fenfluramine treatment.

Line 334: “…implies the absence of pulmonary hypertension (PMID: 334 30689192).” Is this supposed to be a citation?

Line 367: Why is pathogenic capitalized?

Author Response

Thank you for your comments. We apologize for the discrepancies in numbering. Please refer to the pdf version to view the new revisions and to make sure the line #s remain consistent.

Abstract lines 25-28: It is unclear whether the results summarized are from the literature or from the case study.

Lines 25-28 were referring to the case study/Index case. We revised our abstract to clearly summarize the results obtained from the literature and the case report. Please see PAGE 1 LINES 31-35 IN THE EDITED VERSION.

Introduction: Lines 62-63: Statement regarding B1B effect on neurite outgrowth requites a citation.

We added a citation #19 for this statement. See PAGE 2 lines 145.

Introduction Line 108-109: Statement reads: “the heterozygous SCN1B C121W (SCN1B +/C121W) mice have a lower seizure threshold than heterozygous SCN1B null (SCN1B+/-) mice, and this implies a deleterious gain of function mechanism.” This is also included in Table 1. However, this implication is not completely supported by the data cited since the heterozygous missense variant should be compared to wild type littermates, and the same with the heterozygous null. It is unclear how a valid comparison was made between null hets and variant hets, unless the null and variant hets were crossbred, and then reassorted randomly with WT littermates.  Short of this, the support for in vivo gain of function for C121W is weak.

Please see explanations on PAGE 3: 159-174, PAGE 8: 519-527, PAGE 9-10: 623-780 and various changes in cases of Table 2 in edited version.

Introduction: Related to the above comment, only 1 variant (C121W) was tested using a mouse/animal model, all others were tested in vitro. This should be noted as a limitation, since in vivo and in vitro studies often contradict one another.

Explanation added on PAGE 8 paragraph 3 IN EDITED VERSION.

Table 1: Why is there no LOF/GOF designation for the last 3 entries?

Table 2 has been edited to show these changes. Please see the table 2 in the EDITED VERSION on page 6-7.

Introduction lines 129-130: Statement reads: “This implies that SCN1B variants may directly result in TLE.” Were patients in this study screened for variants in other sodium channel genes? If not, this limitation should be mentioned since gene/protein interactions may be involved in such phenotypes. Similarly with other studies that are cited in regards to pathogenic effects of SCN1B variants in different epilepsy syndromes: were other sodium channel genes evaluated in those studies? If not, again this can be cited as a limitation.

Please see new reference added to support the variants. Please see Page 8-9 line 539-545 IN EDITED VERSION.

Lines 208-210: Statement reads: “It is reported that various SCN1B-related developmental and epileptic encephalopathies impact both excitatory and inhibitory neurons, resulting in the increased severity of EIEE52.” This does not seem accurate and/or is confusing. Wouldn’t it be the variants that affect both types of neurons and result in increased severity (not the encephalopathies)? Related to this, wouldn’t most/all SCN1B (and SCN1A) variants cause sodium channel dysfunction in both excitatory and inhibitory neurons? There is also the issue that loss of function of the beta subunit may result in gain of overall channel function (i.e. hyperexcitability). These effects should be distinguished as they are highly relevant to therapeutic strategies. Overall more discussion/explanation of gain vs loss of function effects of variants would strengthen the manuscript.

We have deleted the statement mentioned. Please note on PAGE 8 LINES 517-522 that we mention the loss of function beta subunit resulting in a gain of function mechanism. We also summarized a bit on the overall impact variants have on the two types of neurons on LINES 531-534, however we did not elaborate further.

Lines 228_230 regarding fenfluramine: “It is an amphetamine derivative that primarily affects serotonin neurotransmitter levels”. In fact, although synaptic levels of serotonin may be increased transiently (not overall brain levels), the statement as written disregards the changes in receptor density that accompany the therapeutic use of this drug and that are thought to underlie therapeutic benefits in depression. There are other effects as well, and these all should be briefly reviewed. Special attention should be given to downstream effects such as on global gene transcription and cellular effects such as neurogenesis.

This is a good suggestion, we have elaborated a bit on some of the other mechanisms, see page 10 lines 811-816. We were not able to find a citation regarding the receptor densities.

We did not want to go too much into detail as we feel that this may be beyond the scope of this review article focused on the genetic component (SCN1b) of a specific infantile epilepsy.

Line 276: More information on the initial genetic panel should be provided.

More information was provided about the genetic panel with a full paragraph dedicated to it. Please see PAGE 4 line 233-242 in the edited version.

Comments on the Case Report:

Lines 301-310: It would be interesting to examine possible association between medication changes and EEG changes. Was the patient on different combinations of drugs at the times of the various EEG tests?

Please see Figure 2 on PAGE 5 to see the changes in the medication timeline.

Lines 321-322: “Early infancy age” does not seem like a criterion that would support or justify the use of a medication approved only for older children. It seems like the opposite.

“The term “early infancy age” was used to signify the urgency to treat the patient, in regards to not only controlling daily seizures, reducing episodes of prolonged status epilepticus and PICU admissions, but also to preserve development. Infant should be treated aggressively to preserve their development and control their seizures. It is a desperate measure that we have used a medication not approved this age. We hope this answers the question.

Lines 324-325: How was it determined that the patent foramen ovale was not clinically significant or even related to recurring seizures?

Patent foramen ovale/PFO was determined nonsignificant by cardiology team and it does not cause seizures to our knowledge as direct cause.

Line 335: The statement is made regarding recommendations from cardiology for this patient: “..avoid all drugs contraindicated for patients with Brugada syndrome”. Did this include any drugs that the patient was taking at the time?

We consulted cardiology and followed cardiology recommendation with continuous monitoring and avoiding any medications under PICU team/Cardiology supervision. This was a recommendation to be strictly followed by cardiology supervision. This did not include any drugs that the patient was taking.

Table 2: The data in this table are not presented clearly. Thus, the therapeutic impact of the various treatments, including fenfluramine, cannot be determined. What drugs was the patient on when fenfluramine started? What was the time course of changes in the patient’s ASM regimen after fenfluramine was added? What was the order of weaning? What are the “% reductions” in reference to? Are these subjective? The authors should consider an alternative presentation for Table 2, with addition of more data.

We added figure 2 (PAGE 5) to reflect the time course of ASM regimen changes, when fenfluramine was started, and order of drug weaning. Percentages reflect the reduction in seizure burden.

What level of genetic testing and what other genetic information is available on the patient? Have other ion channel genes been screened?

Invitae Epilepsy panel and add on preliminary genes for epilepsy was performed on urgent basis through inpatient care which included 308 genes for genetic disorders. Other variants of unknown clinical significance such as LMNB2, RBFOX3 and RELN, SCP2 were noncontributory.

Does the patient have any siblings?

One full biological healthy 4years old brother with no neurological concerns and with normal development.

Minor comments:

Throughout: English grammar requires refinement (e.g. lines 71-73 and 78-79).

We have made corrections to the grammar.

There are instances where “patient-first” language is not used (e.g. line 83 should be: “patients who experience sudden death from epilepsy (SUDEP)”.

We have changed the language to reflect patient-centered language.

TLE is defined on line 121 but temporal lobe epilepsy is first mentioned on Line 85 and the acronym should be defined there. Also, the acronym should be used on line 125.

We have corrected this so that the acronym is defined at the first mention of TLE.

Lines 130-131: There seems to be word missing in this sentence. Should this be: “..notably the only de novo SCN1B [VARIANT] previously described”?

Yes, we added the word variant.

LOF is define on Line 134, but is used several times before this including in Table 1. Similar problems are noted with other acronyms as well (e.g. ECHO). Poor organization and lack of systematic use of acronyms detracts from the manuscript.

We have done a revision to address this concern by reviewer to best of our ability.

Lines 142-143: This is not a complete sentence

Please see PAGE 9 589-591 in the EDITED VERSION. We completed the sentence.

Note redundancy between lines 244-245 and 251-252 regarding need for cardiac monitoring during fenfluramine treatment.

We have removed this redundancy by stating it only once.

Line 334: “…implies the absence of pulmonary hypertension (PMID: 334 30689192).” Is this supposed to be a citation?

We have changed this to a reference.

Line 367: Why is pathogenic capitalized?

We have made changes to this.

Reviewer 2 Report

This is a nice review of SCN1B variants and their association with epilepsy, with the goal to outline the background of a clinical decision to treat a pediatric patient with fenfluramine. Structure of voltage-gated sodium channels is discussed, focusing on the beta subunits and the phenotypes of SCN1B variants. GEFS+, focal epilepsy and DEE are briefly presented and medications previously reported to be beneficial in SCN1B-related disorders, namely phenytoin and fenfluramine, are presented. The review part is complete, well-presented and supported by appropriate references.

The case report is a patient with compound heterozygous SCN1B variants. The clinical course and interventions are described in detail. The patient was started on fenfluramine at the age of 8 months on a compassionate basis, and the authors comment on the effectiveness and safety of this intervention.

This report is of great value and adds to the literature on the use of fenfluramine, and at the same time provides a nice review of the SCN1B related disorders.

1. As is always the case with complex epilepsies, this patient has a complicated medication history and had been on several antiseizure medications. Therefore, it would be beneficial for the reader to have a figure summarizing the time-frame of adding and withdrawing medications, preferably with the dose and the number of episodes – perhaps similar to Figure 5 of the first reference of the manuscript, which deals with a similar case. This is also important as after fenfluramine was added, clobazam and topiramate were withdrawn, and valproate, phenobarbital, and cannabidiol (Epidiolex) were added. The series of these interventions are not clear enough in the manuscript. Several questions may be answered with these details: for example, cannabidiol and clobazam are reported to have a synergistic effect; where those co-administered at any time?

The dose of the medications should also be mentioned in the text whenever possible.

2. The abstract should include more details on the case report. For example, the age at fenfluramine initiation is not mentioned.

3. The authors do not discuss the functional aspects (LOF?) of the specific variants of their case.

4. Authors should avoid using brand names (Keppra, Topamax).

Author Response

Thank you for your comments. 

1. We added a new chart to depict the timeline of medication usage (Figure 2).

We added all the doses of medications where applicable.

2. Age of fenfluramine initiation was added to the abstract. We also added a few more details on the case report in the abstract.

3. Functional studies have been performed on the SCN1B p.R85C variant previously, as referenced in Table 1. Although functional studies have not performed on the p.Met1? variant, we have added that this variant was reclassified as likely pathogenic, based on the increased frequency of this variant in affected individuals. 

4. All mention of Keppra and Topamax has been changed to Levetiracetam and Topiramate throughout the paper.

Author Response

Thank you for the comments. 

1. We clarified that the p.Met1? variant was reclassified as likely pathogenic, based on the increased frequency of this variant in affected individuals. Combined with the p.R85C variant (in trans), the index case carries compound heterozygous SCN1B gene variants.

2. We restructured the manuscript to have the case report and the literature review in “results” and added elements to the discussion that are relevant to the literature review.

3. We restructured the paper and added a title 3. Review of the spectrum of associated phenotypes with SCN1b variants.

4. We replaced all mention of EIEE with DEE.

Round 2

Reviewer 1 Report

Prior comment:

Introduction Line 108-109: Statement reads: “the heterozygous SCN1B C121W (SCN1B +/C121W) mice have a lower seizure threshold than heterozygous SCN1B null (SCN1B+/-) mice, and this implies a deleterious gain of function mechanism.” This is also included in Table 1. However, this implication is not completely supported by the data cited since the heterozygous missense variant should be compared to wild type littermates, and the same with the heterozygous null. It is unclear how a valid comparison was made between null hets and variant hets, unless the null and variant hets were crossbred, and then reassorted randomly with WT littermates.  Short of this, the support for in vivo gain of function for C121W is weak.

Author response: Please see explanations on PAGE 3: 159-174, PAGE 8: 519-527, PAGE 9-10: 623-780 and various changes in cases of Table 2 in edited version.

Reviewer response: Lines 159-174 do not address mouse models; lines 519-527 do address the mouse models, but the text is essentially unchanged from the original submitted version; lines 623-780 discuss the null Scn1b mouse, but they do not address the comparison between the KO and knock-in models wherein the C121W variant is described as gain-of-function. Thus overall, the original criticism remains unaddressed. In other words, there appears to be little to no support for the idea that C121W is a GOF variant.

Prior comment:

Introduction: Related to the above comment, only 1 variant (C121W) was tested using a mouse/animal model, all others were tested in vitro. This should be noted as a limitation, since in vivo and in vitro studies often contradict one another.

Author response: Explanation added on PAGE 8 paragraph 3 IN EDITED VERSION.

Reviewer response: The information added to the end of the 4th paragraph on Page 8 (there are no edits tracked in the 3rd paragraph) does not address the fact that few variants have been tested in vivo and thus the in vitro data are largely unvalidated in terms of LOF/GOF. Thus, the initial criticism remains unaddressed. A good place to address this limitation would be after lines 324-326.

Prior comment:

Introduction lines 129-130: Statement reads: “This implies that SCN1B variants may directly result in TLE.” Were patients in this study screened for variants in other sodium channel genes? If not, this limitation should be mentioned since gene/protein interactions may be involved in such phenotypes. Similarly with other studies that are cited in regards to pathogenic effects of SCN1B variants in different epilepsy syndromes: were other sodium channel genes evaluated in those studies? If not, again this can be cited as a limitation.

Author response: Please see new reference added to support the variants. Please see Page 8-9 line 539-545 IN EDITED VERSION.

Reviewer response: The lines referred to do not address the original comment. In sections 3.2.1, 3.2.2 and 3.2.3, information should be provided regarding the full complement of genetic tests performed on the patients in which SCN1B variants were detected. This also holds for the case report: what genetic testing panel was used and how deep was the search for gene variants?

Prior comment:

Line 276: More information on the initial genetic panel should be provided.

Author response: More information was provided about the genetic panel with a full paragraph dedicated to it. Please see PAGE 4 line 233-242 in the edited version.

Reviewer response: Line 233 refers to an “epilepsy gene panel” but no additional information is provided. Thus, the original criticism remains unaddressed.

Prior comment (related to above comment):

What level of genetic testing and what other genetic information is available on the patient? Have other ion channel genes been screened?

Author response: Invitae Epilepsy panel and add on preliminary genes for epilepsy was performed on urgent basis through inpatient care which included 308 genes for genetic disorders. Other variants of unknown clinical significance such as LMNB2, RBFOX3 and RELN, SCP2 were noncontributory.

Reviewer response: The above information should be added to the manuscript including a reference for the Invitae gene panel and information on the variants of unknown significance. How is it known that they were non-contributory?

Prior comment:

Table 2: (…..)  What are the “% reductions” in reference to? Are these subjective? The authors should consider an alternative presentation for Table 2, with addition of more data.

Author response: (…) Percentages reflect the reduction in seizure burden.

Reviewer response:  it remains unclear how reduction in seizure frequency was quantified.  If the percentages are subjective determinations, that should be stated and it should be explained how they were identified and characterized. For example, did parent keep a record? How were the different kinds of seizures recognized?  As it currently stands, it is unclear how these data were collected and how they should be interpreted. Also, in some cases the reduction is stated as a single value whereas in others a range is given. Why is this? For the table to have value, more explanation of all of these issues is needed.

Prior comment:

Does the patient have any siblings?

Author response: One full biological healthy 4years old brother with no neurological concerns and with normal development.

Reviewer comment: This information should be added to the manuscript.

Reviewer 3 Report

In lines 41 and 123, and Table 2 c.136C>T,  c.178C>T, and c.472G>A, please revise "early infantile developmental and epileptic encephalopathy (DEE)" to "developmental and epileptic encephalopathy (DEE)". 

In line 160, please revise "...MR brain..." to "...brain MRI..."
